# Science and Hurling: A Review

**DOI:** 10.3390/sports10080118

**Published:** 2022-07-29

**Authors:** Kieran Collins, Thomas Reilly, Shane Malone, John Keane, Dominic Doran

**Affiliations:** 1The Gaelic Sports Research Centre, Tallaght Campus, Technological University Dublin, D24 FKT9 Dublin, Ireland; shane.malone@tudublin.ie (S.M.); d.a.doran@ljmu.ac.uk (D.D.); 2The Tom Reilly Building, Research Institute for Sport and Exercise Sciences, Liverpool John Moores University, Liverpool L3 5UX, UK; t.p.reilly@ljmu.ac.uk (T.R.); john.keane@mytudublin.ie (J.K.)

**Keywords:** intermittent exercise performance, performance characteristics, running performance, preparatory practices, injury, talent development

## Abstract

Hurling is one of the world’s fastest field sports. Since the last review of science and Gaelic sports in 2008, there has been an increase in sports science provisions across elite and sub-elite cohorts, resulting in increased hurling-specific literature equating to an additional 111 research investigations into the game across all sports science disciplines. The present review aims to provide an updated analysis of the current research on the game and propose recommendations for future research. Overall, intermittent aerobic fitness remains an important physical quality during competition, with a focus on games-based training methodologies within the literature. Within the current review, we provide updated normative data on the running demands, physiological responses, and anthropometric and performance profiles of hurling players. The increased literature across the sport has led to the development of a hurling-specific simulation, that can now be utilised practically in training and research processes for hurling cohorts. Furthermore, the monitoring of internal and external training loads across training and match environments, in addition to response variables such as well-being, appears to have become more prominent, allowing practitioners to design training regimes to achieve optimal dose and response characteristics. Analysing the game from a scientific perspective can allow for more efficient preparatory practices, to meet the specific requirements of players at all age levels. Collaborative research among the various sports science disciplines, is required to identify strategies to reduce the incidence of injury and enhance performance in hurling. The current review provides updated information to coaches and practitioners regarding position-specific physical qualities, and match-play demands that can concurrently support the training process within hurling.

## 1. Introduction

Hurling is an amateur sport native to Ireland. It is one of the national sports governed by the Gaelic Athletic Association (GAA) since its inauguration in 1884 [1,2]. Inter-county (elite) hurling competitions began in 1887, with the premier competition being the All-Ireland Championships [2]. National interest in the sport is significant, illustrated by the capacity crowds in the latter stages of the All-Ireland Championships, with over 82,000 spectators present in Croke Park on All-Ireland finals day. In recent decades, the popularity of Gaelic games has expanded internationally. The All-Ireland finals in 2020 were viewed in over 170 countries via the GAA’s national streaming service GAAGO. Hurling is an intermittent, high-intensity team sport competed by two teams, each having 15 players on a pitch (130–165 m long) at any one time [3,4,5]. During a game, a wide range of offensive and defensive skills are executed at high speed. These range from striking the ball in the air (Figure 1) or on the ground, lifting, catching, and fielding the ball, and hooking or blocking an opponent (Figure 2) when striking the ball [6,7]. Elite hurling consists of inter-county players, whereas sub-elite hurling comprises club-level players. An elite match has a 70-min duration (2 × 35 min halves), with a sub-elite game being 60 min in total (2 × 30 min halves); additional time is at the discretion of the officials [8,9]. The games consist of a goalkeeper and five outfield positional lines (full-back, half-back, midfield, half-forward, and full-forward), with the aim being to outscore the opposition, which results in a game that ebbs and flows in line with match-play actions. O’Brien, Martin, and Bradley [10] observed that the winning team has a higher total shot count (40), shot count from open play (29.5), and shot efficiency from both open play (61%) and dead balls (77%). Hurling is considered to be the oldest of the stick-and-ball games, and is thought to have been the original root of hockey in its various forms [11]. The game has been described as one of the world’s most dynamic and skilled field games [12]. Hurling is played with a small ball and a curved wooden stick. The ball or “sliothar” is composed of cork surrounded by leather, and is similar in size to a hockey ball but with raised ridges and can be propelled up to 160 km·h^−1^ [13]. The goalposts are the same shape as those on a rugby pitch, with the crossbar lower than in rugby and slightly higher than in soccer. To score, the hurler must put the ball over the crossbar with the hurley for one point, or underneath it and into the net for a goal. A goal is the equivalent of three points.

Despite the dynamic nature and spectator appeal of the game, research into hurling has largely lagged behind Gaelic football and other field games [1,2]. Due to recent pressures from within the Gaelic Athletic Association (GAA), the Strategic Vision and Action Plan (2009–2015) identified the need to prioritise the support and development of players and coaches as key to the long-term success and health of Gaelic games. The recognised need for contemporary knowledge to infiltrate the sport has contributed to promoting change. A new working group to put sports science at the heart of Gaelic Games has been formed, with a remit to generate a framework for the delivery of sports science across all stages of the player pathway. The main driver has been to base the preparation of players on a scientific foundation as evidenced in other sports. The present review aims to provide an updated analysis of the current research on the game and propose recommendations for future research.

Where possible, the current review will communicate data per the line of the field. The literature search process was conducted across several search engines such as Pubmed, SPORTDiscus, and Scopus, with key author names and text search terms, such as ‘hurling’ and ‘anthropometric’, performance’, ‘work-rate’, ‘running performance’, ‘physical demands’, ‘training demands’, ‘physiological demands’, ‘small-sided games’, ‘nutrition’, and ‘injury epidemiology’, used to find peer-reviewed investigations pertaining to hurling. The current review will focus on the male hurling population, specifically discussing the elite and sub-elite populations at the adult level. No inferences are made for female players. In this review, an overview of current research is provided and directions for future research are identified. The anthropometric and performance characteristics of players, running demands of the game, game simulations, preparatory practices, and injury are addressed. Where research on hurling is absent, research from Gaelic football is utilised as the games are played on pitches of the same size area, the teams consist of the same number of players, and participants regularly play both sports.

## 2. Anthropometric and Performance Profiles

### 2.1. Anthropometric Profile

Anthropometric characteristics (body mass, stature, skinfold thickness, adiposity (%AT), and fat-free mass (FFM)) of team sports players have been directly related to levels of performance in competitive match-play within the literature [14,15,16,17]. The rationale for determining an athlete’s body composition relates to the regulation of adiposity levels (%AT), and the encouragement of proper weight management to optimise sporting performance [17]. The anthropometric profiles of hurlers have been previously reported [8,15,17,18,19,20], see Table 1. Early research focused on the stature, body mass, and adiposity (body fat percentage (%BF), as it was termed) of elite hurlers [8,19,20,21]. Initially, the stature of elite hurling players was reported to range between 174.5 and 178.7 cm [19,20]. Additionally, the body mass of players in these studies ranged from 73.4 to 83.4 kg [8,19,20]. The reported adiposity levels of players were shown to have increased variance compared to the stature and body mass (9.7–18.4% (%BF)) [8,19,20,21]. Initial anthropometric findings within hurling cohorts were reported without specific reference to positional lines of play. Within practical settings, the identification of positional profiles can be considered key with respect to the development of physically robust players, who can withstand the training and calendar demands of hurling match-play.

The first study to provide normative positional anthropometric profiles was completed by Collins and Colleagues [15], with the stature (181.7 ± 6.2 cm) and body mass (80.6 ± 8.0 kg) of players in line with previous research on elite hurlers [8,19,20]. Additionally, the sum of five skinfolds taken from the biceps, triceps, subscapular, suprailiac, and front thigh was 40.1 ± 5.6 mm. Adiposity (%AT) was derived from the sum of skinfolds using the Durnin and Womersley equation [22] and was lower than previously reported in elite hurlers (12.7 ± 1.8%). The lowest calculations of adiposity (9.7 ± 1%) were completed using the Reilly equation [23].

Specific profiles for all anthropometric characteristics were reported across the positional lines of play (goalkeeper, full-back, half-back, midfield, half-forward, full-forward). Although no significant differences were reported, goalkeepers were taller (184.3 ± 3.7 cm) and heavier (88.7 ± 5.7 kg) than all other positions apart from full-backs (186.6 ± 5.1 cm). The sum of skinfolds ranged from 41.6 ± 9.0 mm for goalkeepers to 41.6 ± 2.7 mm for midfielders to 38.3 ± 5.3 mm for half-backs. Furthermore, %AT ranged from 13.2 ± 3.1% for goalkeepers to 11.8 ± 2.1% for half-backs [14,15]. Practitioners within elite hurling teams now place an increased emphasis on the body composition profiles of their players, with seasonal testing in the form of dual-energy X-ray absorptiometry (DXA) analysis now the common approach to pre-season and in-season anthropometric assessment [17,18,24]. Limitations have been reported in relation to using skinfolds to infer the body fat percentage estimations [25]. It was suggested that the conversion of absolute skinfold thickness to %BF should be reconsidered, as all equations fail to predict %BF relative to the gold standard DXA scan [25,26]. Accordingly, the most up-to-date body composition data for hurlers have been provided from DXA data [17,18]. Hurling players were reported to have a lower level of lean mass (LM; 67.6 vs. 68.3 kg) and fat mass (FM; 12.6 vs. 12.8 kg) compared to Gaelic footballers, with the percentage of fat mass reported as similar for both (14.9 vs. 14.9%) [17]. Following on from this work, seasonal variations in the anthropometric characteristics provided by the DXA scan were reported for elite hurlers [18]. Significant changes were observed in FM and LM within a season in response to temporal changes in training, and competition with respect to elite-level hurlers [18]. As the preparatory practices regarding nutritional considerations, body composition analysis, and training load monitoring improved with the increased prevalence of sports science, the differences in anthropometric characteristics across an elite squad of players have reduced, and are now minimal in nature owing to the increased individualisation of nutritional support for players.

### 2.2. Performance Profiles

Performance profiling of athletes can provide important baseline data for characteristics such as strength, upper and lower limb power, speed, acceleration, and aerobic fitness capacity [15,27,28,29,30,31]. Early research reported aerobic fitness performance as the estimated maximal oxygen uptake (V̇O_2max_: mL·kg^−1^·min^−1^) based on different estimation protocols [8,14,19,20]. The estimated VO_2max_ values ranged from 47.4 ± 7.0 to 58.9 ± 4.8 mL·kg^−1^·min^−1^ [8,14,19,20]. Collins and colleagues [15] provided the first positional aerobic fitness profiles within an elite hurling cohort, with the average estimated VO_2_ reported as being 56.3 ± 2.9 mL·kg^−1^·min^−1^ [15]. Midfielders were observed to have significantly greater (60.1 ± 1.4 mL·kg^−1^·min^−1^) estimated VO_2max_ based on completed multi-stage fitness tests, compared to all other positions—goalkeepers (50.3 ± 2.0 mL·kg^−1^·min^−1^), full-backs (56.2 ± 1.6 mL·kg^−1^·min^−1^), half-backs (57.1 ± 1.8 mL·kg^−1^·min^−1^), half-forwards (56.1 ± 1.9 mL·kg^−1^·min^−1^), and full-forwards (56.0 ± 1.8 mL·kg^−1^·min^−1^). Goalkeepers were also reported to have significantly lower estimated VO_2max_ compared to half-backs.

Recently, the findings reported in hurling and Gaelic football cohorts concerning aerobic fitness measures, have utilised aerobic field-based fitness tests (Yo-Yo IRT1; Yo-Yo IRT2) for the analysis of aerobic performance [21,28,31]. Keane et al. [21] observed that in the Yo-Yo IRT1, elite players completed a greater distance than sub-elite players across all positions. When seasonal variations were analysed in a Gaelic football cohort, significant improvements were reported as the season progressed [28]. Specifically, a 34.9% increase in the distance completed was observed across a season (pre-season, 1174 ± 335 m; in-season, 1567 ± 333 m) [28]. In line with this research, the mean Yo-Yo IRT2 scores reported within a similar elite Gaelic football cohort were 1587 ± 298 m. The Yo-Yo IRT1 of a sub-elite level squad prior to and after a small-sided game (SSG) intervention of 12 SSGs with differing rules across 4 weeks, reported improvements in the Yo-Yo IRT1 from the pre- (1640 ± 276 m) to the post- (1880 ± 177 m) intervention periods [30]. As high-speed running (HSR) capacity is considered an important attribute during hurling match-play, evaluating the intermittent running capacity of players can identify suitable positional roles, while also allowing practitioners to identify players who require specific training plan modifications to improve this capacity [15].

Speed and acceleration capacity are essential attributes related to hurling performance, with numerous decisive moments determined by the ability to produce quick bursts of anaerobic activity [7,32]. Successful performance can be influenced by the ability to accelerate over 20 m, both in offensive and defensive positions to obtain possession of the sliothar [28,30,33,34]. The acceleration profiles reported during elite hurling match-play were shown to be similar to Gaelic football, with players performing one acceleration every 22 s per match (hurling; 189 ± 34, Gaelic football; 184 ± 40) [12,35]. When assessing these characteristics, sprint speed is most often measured in a straight line, typically in the range of 5–40 m sprints, whereas acceleration profiles refer to the shorter distances of 5, 10, and 20 m [14,15,31]. Initial speed data were reported over 10 m with respect to hurlers (1.75 ± 0.03–2.29 ± 0.1 s) [14,19,20]. These times were slower than those reported for English Premier League soccer (1.73 ± 0.08 s) and Australian Rules football (1.70 ± 0.2 s) players at that time [36,37]. Similar to the aerobic fitness profiles above, normative positional profiles for accelerations over 5, 10, and 20 m were reported for the first time within elite hurling [15] The average sprint times for the squad across each distance were 1.00 ± 0.04, 1.86 ± 0.04, and 3.03 ± 0.06 sec for 5, 10, and 20 m, respectively. No significant positional differences were reported, with a range of sprint times at 20 m (3.03–3.04 s), which was reflective of the similarities across positions at an elite level. Indeed, hurlers appeared to be quicker across 5 m and 20 m distances compared to Gaelic footballers upon review of the current data [15,31].

Seasonal variations have been observed in the acceleration profiles of players, with a 7% reduction in 5 m sprint times reported from the pre-season (1.15 ± 0.09 s) to the early in-season phases (1.07 ± 0.07 s) of that year [28]. The progressive increase in training regimens combined with decreased body composition, were likely attributable to the improvements in sprint times across the season considering that increased adiposity can potentially compromise the efficiency of certain sport-specific movements, such as accelerations, sprints, jumps, and prolonged running [1,2,19]. The most recent acceleration profiles reported in the literature observed a sub-elite cohort assessed for acceleration capacity prior to and after a specific SSG intervention [30]. No significant differences were reported in sprint times across 5 m (pre, 1.00 ± 0.44 s; post, 1.00 ± 0.43 s), 10 m (pre, 2.11 ± 0.65 s; post, 2.11 ± 0.77 s), and 20 m (pre, 3.44 ± 0.78; post, 3.33 ± 0.76 s), which was in contrast to the other characteristics reported [30]. These findings potentially suggest that athletes require specific adaptive training, unlike SSG conditioning, to directly improve speed qualities [30].

Explosive power is an additional performance characteristic of considerable importance with respect to intermittent team sports [1,2]. Along with acceleration, sprint, and aerobic capacity, vertical jump ability is a key component of the performance profile of a hurler [7]. Contested aerial duals are frequent in hurling match-play, with the attainment of primary possession being of utmost importance [1,2,3,4,5]. Therefore, the importance of jumping ability and the explosive power capacity to propel body mass upwards appears apparent [1,2,3,4,5,6]. Vertical jump (VJ) or countermovement jump (CMJ) performance has been used as a measure of lower limb power across multiple studies, with differences observed when compared to other intermittent team sports [15,28,31,38].

To further investigate the lower limb power of athletes, certain studies have reported peak power (PP; Watts) and relative peak power (RPP; W·kg^−1^) to investigate jumping ability with respect to individual body mass [15,31]. Previous research has reported the CMJ profiles of hurlers to range between 33.0 ± 2.7 and 56.6 ± 8.0 cm [8,14,19]. The findings reported by Brick and O Donoghue [8] appear significantly higher than typically reported and may suggest potential reliability issues with the jump apparatus used (hurlers, 56.6 cm; Gaelic footballers, 62.2 cm) or the utilisation of an arm swing within the testing procedure. The CMJ and lower limb RPP profiles for hurlers have been reported as well as the mean CMJ performance (47.2 ± 5.1 cm) and RPP (55.4 ± 3.8 W·kg^−1^) [15]. Positional profiles exist, with hurlers reported to be similar to these across positions (full-back 45.2 ± 7.7 cm to full-forward 50.8 ± 5.9 cm). Given the demands of the sport of hurling with its fast-paced end-to-end deliveries, short-distance sprints, and abrasive tackling, it would appear that speed, acceleration, power, and aerobic fitness capacities are attributes that contribute to optimal performance within competition [1,2,3,4,5,6,15,29,30]

### 2.3. Running Performance Profile

Hurling is a high-intensity intermittent field sport, with numerous studies outlining the prevalence of short bursts of high-speed efforts, superimposed on longer periods of active recovery, due to the ebb-and-flow nature of competitive match-play [1,3,4,5,7,30,39,40,41,42,43,44,45,46]. Due to the distance that the ball can travel, the game has a unique profile with scores coming from one and two-pass possessions [47]. Understanding the specific movement patterns and running demands within a sport supports the implementation of appropriate training regimens and preparatory practices [3,4,5,7,35,39,40,41,42,43,44,45,46]. Expanding the knowledge of these specific match-play demands, requires observational studies to be completed utilising a method of internal (heart rate—HR) or external (Global Positioning Systems—GPS) monitoring markers to develop overall team demands, as well as specific positional and temporal profiles with respect to match-play [3,4,5,39,40,41,42,43,44,45,46]. In recent years, there has been a considerable increase in the number of investigations dedicated to the running demands of hurling [3,4,5,7,35,39,40,41,42,43,44,45,46].; see Table 2.

The initial analysis of match-play demands was conducted using a single camera, time-motion analysis system [12]. A single player was observed throughout the match using the camera system, and once the game had been completed, retrospective analysis and coding took place [12]. The player spent 6.6 ± 2.6% of total match time standing stationary, 43.8 ± 4.3% walking, 14.8 ± 4.4% shuffling, 9 ± 2.8% jogging, 7.3 ± 2.5% running/sprinting, 3.3 ± 0.8% in sideways movements, and 15.1 ± 2.5% in backwards movements [12]. On average, there was a transition every 3.7 s between different activities. Of these transitions, the ratio between high-intensity activities to low-intensity activities was approximately 1:5.3 [12]. Technological advancements in GPS technologies, have resulted in a significant increase in investigations completed with reference to match-play demands within hurling [3,4,5,7,35,39,40,41,42,43,44,45,46]. The typical metrics reported, include distance-based metrics at varying speeds (TD (m); HSR (m) over (≥17 km·h^−1^); sprint distance (SD) (m) over (≥22 km·h^−1^)), maximal-intensity actions (accelerations, decelerations, maximal sprints), and metabolic estimations (average metabolic power, high metabolic load distance, energy expenditure estimations) [3,4,5,7,35,39,40,41,42,43,44,45,46,49,50,51,52,53,54].

The first investigation designed to specifically determine positional and temporal profiles within elite hurling, was completed by Collins and colleagues [3]. The Match-play running performance was initially reported across a multitude of variables (TD, 7617 ± 1219 m; HSR, 1134 ± 358 m; SD, 319 ± 129 m; accelerations, 189 ± 34; and maximal speed, 29.6 ± 2.2 km·h^−1^). Specific positional demands were evident across total, and high-speed work completed during a game. Midfielders covered significantly more TD (8999 ± 676 m) and HSR (1571 ± 371 m) compared to all other positions (full-back, 6548 ± 786 m, 880 ± 204 m; half-back, 8046 ± 686 m; half-forward, 7975 ± 845 m, 1249 ± 262 m; full-forward, 6530 ± 1112 m) suggesting that the implementation of detailed training prescriptions replicating the positional requirements were warranted within the sport [3]. Additionally, the HSR average reported (1134 + 358 m) was 39% lower than Gaelic football (1695 ± 503 m) [52,55]. Interestingly, decrements in HSR were observed across time with differences between the first (330 ± 120 m) and second quarters (271 ± 107 m) evident [3]. Additionally, post-half-time, there was an observed increase from the second to third quarters (278 ± 118 m), followed by a significant decrease between the third and fourth quarters (255 ± 108 m), as the game progressed towards the conclusion [3]. Investigators have suggested the role of transient fatigue to be a pronounced factor in relation to the decrements in running performance [50,51,52,53,54]. Whether by specific half-time strategies or some form of nutritional intervention during match-play, it appears that strategies to mitigate the significant levels of fatigue induced by the intermittent nature of the game should be considered [55,56].

Additional research has investigated the maximal running intensities [43,52] specific sprinting profiles [5], positional and temporal running profiles [5,42,44,46], and the decrements in running performance across quarters [40,41,42] within elite hurling match-play. The identification of maximal running intensities within elite hurling match-play with respect to positions provided normative data pertaining to the peak intensity periods of match-play across different durations (1 min–10 min intervals) [43]. Using a rolling average method, the calculated maximal relative demands (RTD) (184 ± 21 m·min^−1^), RHSR (51 ± 13 m·min^−1^), and SD (42 ± 10 m·min^−1^) were reported respective to elite hurling match-play. The reporting of smaller duration intervals provided a more reflective profile of the maximal intensity periods compared to full games or quarters. The range of maximal-intensity periods across positions for 1 min of match-play highlighted the relative consistency of greater demands across the transitional lines of play; TD (195 ± 21 m (half-back) to 167 ± 15 m (half-forward and full-forward)), HSR (59 ± 12 m (half-back) to 45 ± 13 m (full-back)), and SD (45 ± 9 m (half-forward) to 39 ± 12 (full-forward)] [45]. Normative positional data specific to peak running intensity across different durations (1–10 min periods) can support practitioners in preparing for worst-case match-play scenarios by facilitating the design of specific conditioning drills exposing players to referenced demands. [43] Subsequently, the typical sprint profiles of elite hurlers were reported for match-play [5]. Total SD (>22 km·h^−1^), number of sprints (<20 m>), and relative speed thresholds (<80, 80–90, >90%) were also reported for full match-play samples. The average total number of sprints during match-play was 22.2 ± 6.8, accumulating to 415 ± 140 m SD. Additionally, the number of sprints completed at <20 m (14.0 ± 4.7) was greater than that at > 20 m (8.1 ± 3.6). Furthermore, the number of sprints completed at < 80% (10.6 ± 4.3) was greater than that at 80–90% (8.2 ± 3.6), with the least number of sprints completed at >90% (3.4 ± 2.4) [5]. Players appear to reach peak speeds fewer times than average–high speeds during games, whereas a temporal decrease across all sprint variables was reported with respect to halves of match play. These findings provide normative positional sprint profiles for hurlers during match-play that can be replicated in training sessions to prepare players appropriately for in-game sprint activities [4,5]. Temporal and positional differences in match-play running demands [5,44,45] and running decrements across quarters [40,42] were reported, again focusing on the elite level of play.

The metabolic power approach to estimate the high-intensity demands and energy expenditure during match-play has been utilised within hurling match-play [44]. The findings showed the averages for metabolic power, HMLD, TD, and HSR reported for elite players were 8.9 ± 1.6 W·kg^−1^, 1457 ± 349 m, 7506 ± 1364 m, and 1169 ± 260 m, respectively [46]. The average metabolic demand of hurling match-play was reported to be lower than both Gaelic football (9.5–12.5 W·kg^−1^) [51] and Australian Rules football (9.2–10.9 W·kg^−1^) [57]. This was the first attempt to include the novel metabolic power approach in relation to hurling match-play. As the approach has recently been proposed as ecologically valid for Gaelic football [58], practitioners may consider utilising the approach to further understand the energetic demands of match-play. The inclusion of HR as an objective internal intensity marker, was a further attempt to analyse the overall demand profiles of elite hurlers. Despite the intermittent nature of hurling, heart rate response provides a useful measure of physiological strain, and has been used in other field sports [59,60,61]. During elite hurling match-play the mean heart rate was found to be 83 ± 3 %HR_max_ (163 ± 7 b·min^−1^) with no observed variation between the halves. Young et al. [46] observed a mean HR response of 165 ± 9 b·min^−1^ and a peak HR of 188 ± 9 b·min^−1^ during match-play. The cardiovascular strain imposed during match-play is relatively high and does not fluctuate greatly.

The average TD and HSR (>17 km·h^−1^) reported for elite (TD, 7689 ± 1270 m; HSR, 1174 ± 374 m) and sub-elite (TD, 6574 ± 1205 m; HSR, 868 ± 319 m) players highlighted a greater capacity for HSR within the elite group [39], with elite players completing greater relative high-speed running (RHSR) (17 ± 5 m·min^−1^) compared to sub-elite (15 ± 5 m·min^−1^) players [39]. Elite players also achieved significantly higher maximal speeds (29.8 ± 2.1 km·h^−1^) compared to sub-elite (29.1 ± 1.9 km·h^−1^) players. These findings provided an early indication that differences were present between each level of play. Interestingly, since the completion of the above study, only one additional investigation has directly compared elite and sub-elite match-play running demands [45], whereas the remainder of the work has focused exclusively on the elite level. Elite hurlers were shown to complete greater RTD than sub-elite (118 ± 9 m·min^−1^ vs. 93 ± 16 m·min^−1^) hurlers. Additionally, no differences were reported between levels for RHSR (2.9 ± 1.1 m·min^−1^ vs. 3.3 ± 0.4 m·min^−1^) and relative sprinting (0.24 ± 0.20 vs. 0.27 ± 0.03 m·min^−1^) [47]. It is important to note that the speed thresholds were different for HSR (HSR, 17 km.h^−1^ vs. 19.8 km.h^−1^) and sprinting (SD, 22.0 km.h^−1^ vs. 25.2 km.h^−1^) [3,39,42,44,49,50,51,52,53,54]. To conclude, to advance stakeholder understanding within Hurling and Gaelic games, practitioners need to be able to effectively communicate valid and reliable running performance measures to coaches and key stakeholders through simple reporting strategies, visualization techniques, and verbal communication where appropriate within the training process.

## 3. Simulated Match-Play

The range of factors that affect performance during hurling match-play, creates a challenge for a research scientist to identify the effectiveness of a training intervention or ergogenic aid. To address this challenge, scientists have constructed simulations that aim to replicate the physiological, and running demands of their respective sports. The simulations have ranged from treadmill running to shuttle running [56,62,63,64,65,66,67,68]. A reliable and reproducible multidirectional hurling simulation protocol was designed to simulate the work rate of a typical competitive elite hurling game [69,70]. The protocol has been used to evaluate the training load [71] and the effects of carbohydrates and caffeine on performance [72]. The hurling simulation protocol elicits a mean VO_2_ of 48.1 ± 3.7 mL·kg^−1^·min^−1^, which represents 84 ± 7% VO_2max_ [69,70,71]. During hurling match-play, a heart-rate response of 162 ± 15 beats·min^−1^ (83% HRmax) has been observed [35]. Utilising an estimated HR-VO_2_ regression model, this represents a physiological cost of 70–85% of VO_2max_. The VO_2_ peak attained during the protocol was 56 ± 5.1 mL·kg^−1^·min^−1^. The mean respiratory exchange ratio was 0.82 ± 0.04 with a global energy cost of 5.7 ± 0.8 MJ. Carbohydrate oxidation was 1.9 ± 0.8 g.min^−1^, which elicits a total carbohydrate utilisation of 137 ± 58 g. The rate of fat oxidation was 1.1 ± 0.3 g.min^−1^. The physiological responses are analogous to elite match-play. During the hurling simulation protocol, participants covered 110 ± 2 m·min^−1^, compared to the 109 ± 18 m·min^−1^ observed during elite hurling match-play [69,70]. The HSR distance of the protocol is 19 ± 2 m·min^−1^, which is higher than observed during match-play at 16 ± 5 m·min^−1^ [69,70]. Although the hurling simulation protocol is not a test of any functional capacity, the current data identify that it can be used as a practical and economical field-based simulation.

## 4. Preparatory Practices

In hurling, rapid acceleration and sprinting ability are important to allow hurlers to reach the ball before the opposition [1,2,3,4,5,73]. Plyometrics training for hurling observed improvements in repeated sprints with fewer decrements in performance, compared with baseline scores following a 10-week intervention [74]. High-intensity accelerations are critical in many of the encounters that arise when playing hurling, and the ability to move quickly and repeat the action are important components in the preparatory regimen of hurlers. Byrne et al. [6] investigated a combination of a plyometric exercise and an explosive activity such as a sprint. The authors observed an immediate decline in CMJ measures due to acute muscle fatigue and super compensation augments maximum lower limb strength after 7 days of recovery. Well-developed lower-body strength (relative to body weight), repeated-sprint ability (RSA; s), and speed (s) are associated with better tolerance to higher workloads, and reduced risk of injury in hurling players [29]. Strength plyometric and power-based training within hurling should be appropriately periodised and, tailored by the strength and conditioning coach, based on the individual needs of players. Coaches may develop programs that are reflective of players current capacity in addition to the needs and requirements of training and competitive match-play.

There has been an increased interest in investigating the hurling training process [75,76,77,78]. A multitude of literature has been published related to the games-based approach through conditioned games, SSGs, and specific SSG interventions [30,79,80,81,82,83]. SSGs are now commonly used as a method of conditioning players within team sports including hurling [30,79,80,81,82,83]. These games take place on condensed pitch dimensions, allowing coaches the opportunity to develop both technical and tactical proficiency within players. Furthermore, these games represent a concurrent method of training allowing for the simultaneous development of aerobic and anaerobic fitness characteristics [30,79]. Practically, hurling coaches face the challenge of improving athletic and physiological performance without encroaching on the technical and tactical side of the training process [79,80,81,82,83]. Therefore, the concurrent characteristics of SSGs have resulted in these games representing a major component of hurling training [79,80,81,82,83]. During SSGs, players’ heart rates can exceed an intensity deemed high enough to promote aerobic capacity development (90–95% of maximal heart rate; HR_max_) [77,84]. Investigations of SSGs within hurling have considered a number of prescriptive variables such as player numbers, rule changes, competitive nature, game type, work-to-rest ratio, and pitch size [77,78,79,80,81,82,83], with all shown to alter the physical and physiological demands placed on players. Overall, SSGs have been shown to stimulate physiological and workload intensities similar to those of actual match-play, improving physiological performance capacities [30,82,83]. From a preparatory perspective external load should be monitored across either time on feet or GPS based technologies. It may be suggested that practitioners make use of objective internal load measures {HR monitoring} such as time spent above a percentage of HR_max_ or iTRIMP given that this objective measure has been shown to have a direct association with improvements in physiological markers across training periods {75,77}. Furthermore, objective internal load measures have been shown to have a stronger correlation with changes in physiological and physical performance measures when compared to the more commonly used session rating of perceived exertion (sRPE) [76,77,78]. Although an increased understanding is available, coaches have been shown to employ their own personalised take on games-based methods within the practical setting [85]. Coaches need to remember that although SSGs have been shown to provide an appropriate physical and physiological adaptation across hurling cohorts, they need to explicitly outline the target tactical concepts (tactical problems, tactical themes, principles of play). This ensures players are clear on the purpose of the drills to maximise the concurrent adaptation across the hurling training process.

Hurlers partake in individual and group training sessions, that encompass a systematic and multi-faceted approach to fitness training [86]. The energetic demands of training and competition require that hurlers consume a well-balanced diet. There are limited data on the nutritional status of players of Gaelic games [86,87,88,89,90]. An enhanced understanding of the physiology of players, has allowed the cross-pollination of information about the nutritional needs and dietary behavior to support optimal adaptation, recovery, and performance [1,2]. The preparatory requirements for hurling, and the demands of the game mean that adequate fuelling is essential to optimising performance [1,2]. Elite hurlers reported a mean energy intake of 2651 kcal·day^−1^ comprising 53 ± 4% carbohydrates, 31 ± 4% fat, and 16 ± 2% protein [89]. The authors concluded that the energy consumed was inadequate to meet the preparatory demands of the game. Keane et al., [72] investigated the effects of ergogenic aids, and identified that the co-ingestion of carbohydrates and caffeine reduces the decrements in repeated-sprint performance compared to the placebo in a hurling population. As professional support and the knowledge base related to hurling teams grows, it is unclear whether the information has been put into practice [86,88].

## 5. Injury Profile

Given the high-intensity nature of hurling where full contact is permitted in relation to tackles, aerial duels, and intermittent battles for possession, it is not surprising that injury incidence is common [3,4,5,75,91]. In addition, to the nature of match-play, the preparation practices and time allocated to training now mirror professional levels [3,4,5,72,87], leading to the potential for the mismanagement of training loads, which could result in injury [92,93,94,95]. Recent literature has made significant efforts to further the understanding of injury incidence in hurling and also examine the mechanisms that could potentially reduce injury occurrence in the future [13,94,95,96,97,98,99,100]. To provide some context to the injury situation in Gaelic games, Roe and colleagues [97] investigated the associated financial expenses and claims related to injuries in hurling and Gaelic football [97]. Between 2007 and 2014, EUR 64,733,597 was allocated to 58,038 injury claims. Registered teams had annual claim frequencies of 0.36, with average claim values of EUR 1157.4 (±192 81). Across this longitudinal study, claims for Gaelic football-related injuries (5395.38 ± 1813.74) were consistently greater than hurling (1859.38 ± 580.12), whereas mean adult claims (6217.88 ± 1762.77) were greater than youth claims (1036.38 ± 520.14) [97]. The epidemiology of injuries in hurling has been presented in multiple studies [13,91,96,100]. Blake and colleagues reported mean incidence rates of 76.7 injuries per 1000 h during match-play and 3.87 per 1000 h in training across three consecutive seasons [94]. The significant increase in match-play injury incidence remained apparent in the subsequent literature. Lower limb injuries were the most prominent classification of injury across the three seasons (68–70.5%), whereas upper limb incidence rates were lower (16.6–24%) [94]. Muscle injuries were the most frequent (35.5–42.2%), with hamstring injuries the most frequently reported (15.7–16.5%).

Similar findings for injury incidence were reported for 127 players across 34 weeks of a season [13]. In total, 204 injuries were reported with 104 players reporting an injury. In line with previous research, injury incidence in match-play (102.5 per 1000 h) was 19 times greater than in training (5.3 per 1000 h). The average reported weekly injury incidence was 13.9%, where the majority were new accrued injuries (83.3%) and a very high number of these were acute (80.9%) [13]. Within this study, muscle injuries were again the most frequent (42.2%), with 70.1% and 16.5% of these being lower limb and hamstring injuries, respectively. Of the other injuries, 7.4% were fractures and there were three eye injuries, and one concussion reported [13]. An additional interesting finding reported was that 14.6% of all injuries were recurrent incidents. Further research published by Blake and colleagues [96] recorded 1030 injuries across 2007–2011 seasons, with 1.2 injuries per player sustained by 71% of players. The injury incidence rate was 2.99 per 1000 training hours and 61.75 per 1000 match hours. Direct player-to-player contact was recorded in 38.6% injuries, with sprinting (24.5%) and landing (13.7%) the next most common injury mechanisms respectively. The median time absent from training or games, where the player was able to return in the same season, was 12 days. The majority (68.3%) of injuries occurred in the lower limbs, with 18.6% in the upper limbs. The distribution of injury type was significantly different between upper and lower extremities: fractures (upper 36.1%, lower 1.5%), muscle strain (upper 5.2%, lower 45.8%). Given the rate of recurrent injuries within hurling players need to return to full participation post the completion of a sufficiently progressed return-to-play protocols that encompass both strength and running based components [91,95]. In comparison, with other sports, the injury incidence in hurling match-play was lower than in rugby league, but higher than in rugby union, soccer, Australian football, and Gaelic football [101,102,103,104]. Interestingly, compared to its corresponding national sport, the incidence rates in Gaelic football match-play were significantly lower (51.2–61.2 per 1000 h) than in hurling, whereas the training incidence was similar in both (5.5–5.8 per 1000 vs. 5.3 per 1000) [105,106].

Following this, a prospective study monitored 696 players across 21 teams over 4 years, with 560 lower limb injuries reported [99]. Similarly, injury incidence in match-play was significantly greater (37.6 per 1000) than in training (2.1 per 1000). This study reported the hamstring as the most frequent site of injury (23.6%) and interestingly reported sprinting as the most common mechanism of injury (34.1%) [99]. Overall, there appears to be a significant number of injuries pertaining to hurling match-play. The literature has suggested that inadequate lower-body conditioning may be a predisposing factor to the incidence rates [13,94,95,96,97,98,99,100]. Furthermore, the data suggest that further consideration may be required when designing a training regimen, as the high frequency of soft tissue injuries potentially reflects poor training load management and fatigue-based injury risk. Coherent with the theme of this review, future research may look to investigate the effects of the transition from sub-elite- to elite-level match-play on injury incidence and rehabilitation strategies [93]. As research illustrates a significant disparity between running demands, this could result in an increased risk of injury if the transitional period is not planned and implemented properly. To improve the training process and additional rehabilitation modalities associated with the return to play in Gaelic sports, Roe and colleagues [95] proposed a six-stage operational framework, that is designed to take the individual needs of each player into consideration, by assessing numerous factors when managing the injury risk [95]. The framework allows practitioners to take a six-stage approach to prepare players for competition by assessing the injury trends, the specific sport’s demands, and the profiles of the players (previous injuries, most common injuries, stressors, sporting demands, physical qualities, etc.), it is hoped that by understanding and applying these six stages that hurling and Gaelic sports practitioners will actively reduce the likelihood of injuries in future seasons of Hurling [95].

## 6. Future Directions

Applied sports scientists tend to draw from the general body of scientific knowledge. It is prudent that information based on well-controlled scientific studies that can support applied practices is disseminated across the game. Future research directions should consider the issues outlined below.

Overall, the literature focused on the anthropometric profiling of hurlers is limited and requires some focus from research groups. Of interest are the alterations in anthropometric data across seasonal periods, and also the differentials between the elite and sub-elite cohorts across these performance data.Literature reporting the performance profiles remains limited within a hurling context, with an even greater paucity of research completed in relation to the sub-elite level.Although the normative data provided from GPS and running demand investigations are fundamental for improvements in training programme design, conditioning drill implementation, and overall preparatory practices, in relation to intermittent team sports, numerous contextual factors have been identified as potential limitations to consider when reviewing data. Referenced examples of such limitations include pitch size, match location, level of competition, opposition quality, match importance, weather, and scoreline. Aside from playing position and match quarter, it is not known how running performance in hurling is influenced by contextual factors.Although the concept of understanding running demands has been extensively investigated to date, there appears significant scope to investigate the energetic demands associated with hurling match-play with respect to positions, halves and quarters.A large body of research currently exists relating to intermittent exercise within the framework of field games, most notably soccer. Questions remain as to the biochemistry of intermittent exercise specific to hurling. The game of hurling is played over a shorter duration, and on a larger pitch with increased numbers and a direct comparison may not be appropriate.The contact time between coach and player is considerably less in hurling compared to other professional sports. In order to maximise training times, the appropriateness of training methodology to develop not only physiological characteristics but also cognitive skills requires investigation.Nutritionists are becoming an integral part of the support staff for elite teams. However, there is no evidence that nutritional practices conducive to optimal sporting performance are being followed. Due to the intermittent exercise pattern of hurling, nutritional supplementation, such as fluid, carbohydrate, and creatinine intake, during training and competition may be beneficial; further research needs to be conducted to determine its importance in hurling.There is currently no information comparing the characteristics and key markers in the development pathways of players participating in county and provincial development squads. Such information could support the selection and development of elite players.The study of expertise and expert performance in hurling offers a unique opportunity that may help promote an understanding of the factors that constrain achievement, and the extent to which these may be overcome by systematic engagement in practice and training.The game of hurling has not been the subject of any detailed biomechanical investigation. There are many features of the game that are amenable to biomechanical treatment and there are many opportunities for biomechanists to contribute to the science of hurling. Biomechanical modelling techniques could help in understanding the underlying mechanisms of hurling skills and their performance.

## 7. Conclusions

In conclusion, the need to apply sports science to the game of hurling has been recognised. The preparatory practices, match-play demands, performance profiles, and injury incidences, are in major respects comparable to those of professional athletes. An enhancement of the anthropometric and performance profiles of players is observable, with lower adiposity and higher levels of aerobic/anaerobic fitness evident. The quest for a formula for peaking is especially complex in team sports, where some individuals may be unable to tolerate high training loads and succumb to injury. Overload injuries are a recognised hazard in young players unless appropriate structures are in place to support them through their pathways. Well-developed lower-body strength, repeated-sprint ability, and speed are associated with better tolerance to higher workloads, and reduced risk of injury in hurling. The application of scientific principles to the development of players can help nurture and produce high performers. Of paramount concern is that activities follow best practice guidelines, and are based on up-to-date scientific principles. A multi- and inter-disciplinary support team needs to be an effective and positive influence on the sporting environment and considerate of the elite and sub-elite responsibilities of the players. The support team’s practices need to be informed by existing evidence. The review allows practitioners to embed sports science into existing player pathways specific to age groups.

In this review, we have examined the available research in hurling, and provided a framework for which future research can be based. It is prudent that information based on well-controlled scientific studies that can support applied practices infiltrate the game, and further the understanding of the mechanisms by which players can succeed within the game. The possibilities apply to sub-elite and elite players alike. The current work has been important in providing a significant body of material that can be of importance to both applied practitioners, coaches, and the scientific community.

## Figures and Tables

**Figure 1 sports-10-00118-f001:**
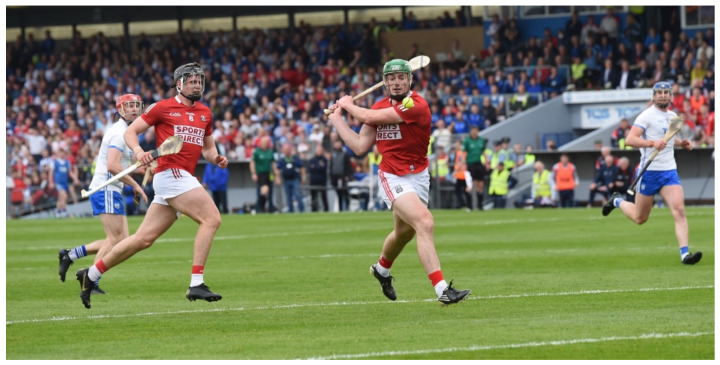
A hurler, striking the sliothar (pic courtesy of G. Hatchell).

**Figure 2 sports-10-00118-f002:**
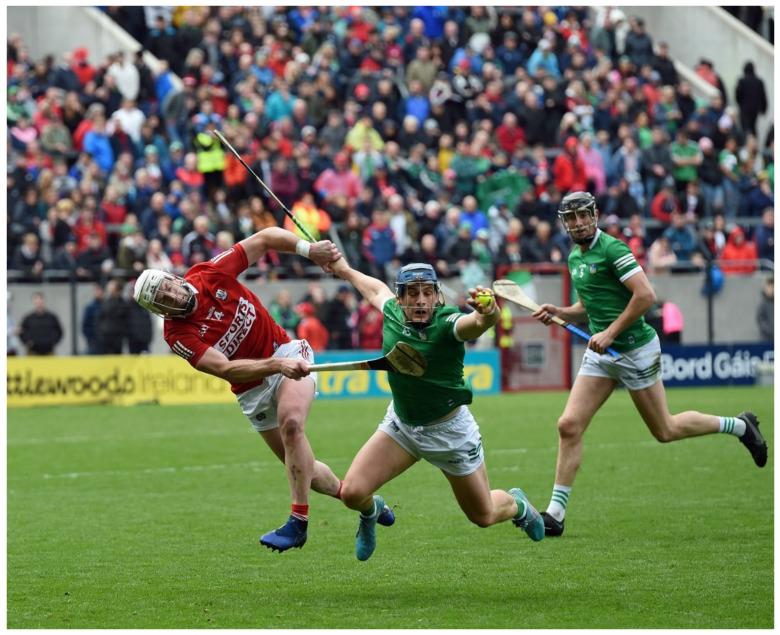
A hurler, catching a sliothar while being challenged by an opponent (pic courtesy of G. Hatchell).

**Table 1 sports-10-00118-t001:** The ages and anthropometric and performance (±SD) profiles for previous hurling studies are presented.

	Doran et al. [19]	Keane et al. [21]	Doran et al. [19]	McIntyre, [20]	Collins et al. [14]	Collins et al. [15]	Keane et al. [21]
	(*n* = 14)	(*n* = 81)	(*n* = 7)	(*n* = 29)	(*n* = 23)	(*n* = 41)	(*n* = 81)
**Level**	Sub-elite	Sub-elite	Elite	Elite	Elite	Elite	Elite
**Age (years)**	26.1 ± 4.0	25.3 ± 5.2		24 ± 5	23 ± 3.4	25 ± 4	24.8 ± 3.6
**Height (m)**	1.77 ± 0.06	182 ± 6.4	1.74 ± 0.05	1.77 ± 0.06	1.83 ± 0.06	1.82 ± 0.06	179.6 ± 3.7
**Body Mass (kg)**	73.8 ± 8.2	80.4 ± 7.5	73.4 ± 7.7	83 ± 9	81.2 ± 8	80.6 ± 7.5	84.1 ± 6.7
**Body Mass Index (kg·m^2^)**	23.6 ± 0.9		24.2 ± 1.1	26.5 ± 1	24.3 ± 0.9		
**Adiposity Tissue (%)**	14.1 ± 3.1	11.1 ± 1.4	13.1 ± 1.4	18.4 ± 3	12.4 ± 2.1	12.7 ± 1.8	9.7 ± 1
**5 m sprint (s)**	-	1.21 ± 0.3	-		0.99 ± 0.04	1 ± 0.03	1.1 ± 0.06
**10 m sprint (s)**	1.78 ± 0.08	1.93 ± 0.3	1.94 ± 0.12		1.77 ± 0.04	1.76 ± 0.04	1.81 ± 0.05
**20 m sprint (s)**		3.2 ± 0.3			3.03 ± 0.07	3.03 ± 0.06	3.08 ± 0.12
**30 m sprint (s)**	4.43 ± 0.17		4.72 ± 0.35		-	-	
**Vertical Jump (cm)**	-	27.9 ± 7.8	-		47.3 ± 6.3	47.2 ± 5.1	45.1 ± 5.1
**Broad jump (cm)**	-	-	-		2.54 ± 0.2	-	

m = meters; kg = kilograms; s = seconds; cm = centimeters; kg·m^2^ = kilogram per meter squared; % = percentage.

**Table 2 sports-10-00118-t002:** The running performance profile of hurling.

**Collins et al.** [3]	4 Hz	VXSport,New Zealand	Elite Seniors	*n* = 94	Quarters	TD (m)	7617 ± 1219	Positional profiles evident with midfielders undertaking the highest volume of work, followed by half-forward and half-back lines. A decrease in HSR distance appears to occur throughout the game, particularly at the latter stages of each half. The positions that completed the highest volume of work also possessed the highest performance decrement.
						TD (m·min^−1^)	109 ± 17
						HSR (m) (≥17 km·h^−1^)	1134 ± 358
						HSR (m·min^−1^) (≥17 km·h^−1^)	16 ± 5
						SD (m) (≥22 km·h^−1^)	319 ± 129
						SD (m·min^−1^) (≥22 km·h^−1^)	5 ± 2
						Max Velocity (km·h^−1^)	29.6 ± 2.2
						Accelerations (n)	189 ± 34
**Young et al.** [45]	5 Hz	SPI Pro, GPSports, Australia	Elite + Sub-Elite Seniors	*n* = 192	Halves	TD (m/min)	E—118 ± 9SE—93 ± 16	Elite level hurlers performed a greater relative TD and TD at walking speeds. Elite players covered a lower TD at running speed than sub-elite. Hurlers at both levels perform sprints over short distances. Temporal differences reported between halves at both levels for relative TD, TD at walking, jogging and HSR speeds.
						HSR (m·min^−1^) (≥19.8 km·h^−1^)	E—2.9 ± 1.1SE—3.3 ± 0.4
						SD (m·min^−1^) (≥25.2 km·h^−1^)	E—0.24 ± 0.20SE—0.27 ± 0.03
**Young et al.** [46]	10 Hz	STATSports, Viper,Northern Ireland	Elite U-21	*Total data points not reported*	Halves	TD (m)	6688 ± 942	At U-21 level, HSR and sprint distance ac-counted for 10% and 4% respectively of the TD covered during match-play. Performance decrements were observed between halves for TD, TD (m/min) and HSR. Players in the full back line covered less TD compared to half backs, midfielders, and half forwards. The commonality among the middle three positions (half backs, midfielders, and half for-wards) emphasizes the need for players to be able to complete the same running performance.
						TD (m·min^−1^)	112 ± 16
						HSR (m) (≥17 km·h^−1^)	661 ± 203
						HSR (m·min^−1^) (≥17 km·h^−1^)	11 ± 3
						SD (m) (≥22 km·h^−1^)	274 ± 111
						SD (m·min^−1^) (≥22 km·h^−1^)	5 ± 2
						Max Velocity (km·h^−1^)	29.1 ± 1.9
**Young et al.** [41]	10 Hz	STATSports, Viper,Northern Ireland	Elite Seniors	*n* = 206	Halves	TD (m)	7807 ± 1094	Between-position differences existed for TD, TD (m·min^−1^), HSR, and SD. Transitional lines of play cover the most distance across higher speeds. Between-half decrements were trivial or small (47 m and 16 m decrements in the 2nd half for HSR and SD, respectively).
						TD (m·min^−1^)	112 ± 16
						HSR (m) (≥17 km·h^−1^)	851 ± 307
						HSR (m·min^−1^) (≥17 km·h^−1^)	12 ± 4
						SD (m) (≥22 km·h^−1^)	340 ± 109
						SD (m·min^−1^) (≥22 km·h^−1^)	5 ± 2
						Max Velocity (km·h^−1^)	30.3 ± 1.8
**Young et al.** [44]	10 Hz	STATSports, Viper,Northern Ireland	Elite Seniors	*n* = 250	Halves	TD (m)	7506 ± 1364	Positional differences shown in metabolic power variables; half-backs, midfielders and half-forwards have increased activity profiles compared to other positions. This is consistent with other locomotion metrics (TD, HSR, etc.) reported previously. Also similar to other locomotion metrics, metabolic power variables, such as HMLD, are subject to fatigue with temporal decrements in performance reported across halves (−96 m).
						TD (m·min^−1^)	107 ± 20
						HSR (m) (≥17 km·h^−1^)	1169 ± 260
						HSR (m·min-1) (≥17 km·h^−1^)	17 ± 4
						SD (m) (≥22 km·h^−1^)	350 ± 93
						SD (m·min-1) (≥22 km·h^−1^)	5 ± 1
						Max Velocity (km·h^−1^)	29.1 ± 2.1
						Accelerations (n)	126 ± 25
						HMLD (m) (≥25 W·kg^−1^)	1457 ± 349
**Young et al.** [43]	10 Hz	STATSports, Viper,Northern Ireland	Elite Seniors	*n* = 230	Periods of gameduration: 1–10 min duration	TD (m)	7358 ± 1085	Worst-case-scenario study—the maximal relative running intensity was 184 ± 21 m·min^−1^. Half-backs, midfielders, and half-forwards completed higher peak TD and HSR intensities compared to full-backs and full-forwards. Higher distances were covered in 1 min; 1 and 2 min; and 1-, 2-, and 3-min durations for relative TD, HSR, and SD, respectively, compared with the 10 min rolling average duration.
						TD (m·min^−1^)	105 ± 16
						HSR (m) (≥17 km·h^−1^)	759 ± 206
					Full game data shown to the right	HSR (m·min^−1^) (≥17 km·h^−1^)	11 ± 3
						SD (m) (≥22 km·h^−1^)	486 ± 127
						SD (m·min^−1^) (≥22 km·h^−1^)	7 ± 2
**Young et al.** [42]	10 Hz	STATSports, Apex,Northern Ireland	Elite Seniors	*Total data points not reported*	Quarters	TD (m)	7853 ± 1124	Temporal decrements between quarters for total distance, HSR, and HMLD. Quarter 1 was the most demanding with greater HSR and HMLD completed compared to all other quarters (Q2–Q4). Position-specific decrements in running performances between quarters were observed in total distance, HSR, and HMLD. Interestingly, there was no difference in the total sprint distance and the number of sprints between quarters in any position.
						TD (m·min^−1^)	112 ± 16
						HSR (m) (≥17 km·h^−1^)	893 ± 270
						HSR (m·min^−1^) (≥17 km·h^−1^)	13 ± 4
						SD (m) (≥22 km·h^−1^)	391 ± 131
						SD (m·min^−1^) (≥22 km·h^−1^)	6 ± 2
						HMLD (m) (≥25 W·kg^−1^)	1607 ± 338
**Young et al.** [48]	10 Hz	STATSports, Viper,Northern Ireland	Elite U-17	*Total data points not reported*	Halves	TD (m)	6483 ± 1145	At U-17 level, HSR and sprint distance accounted for 9% and 4%, respectively, of the TD covered during match-play. Running performances for full games were lower than elite senior hurlers but similar to U-21 level. Temporal decrements in performance were shown across halves for TD, TD (m·min^−1^), HSR, and SD. Positional profiles were reported similar to adult levels; half-backs, midfielders and half-forwards covered more TD, TD (m·min^−1^), HSR, and SD compared to full-backs and full-forwards.
						TD (m·min^−1^)	108 ± 19
						HSR (m) (≥17 km·h^−1^)	583 ± 215
						HSR (m·min^−1^) (≥17 km·h^−1^)	10 ± 4
						SD (m) (≥22 km·h^−1^)	272 ± 77
						SD (m·min^−1^) (≥22 km·h^−1^)	5 ± 1
						Max Velocity (km·h^−1^)	28.1 ± 2.9
**Egan et al.** [7]	10 Hz	STATSports,Apex GNSS,Northern Ireland	Elite Seniors	*Total data points not reported*	Full game	TD (m)	NL—7808 ± 1234	
							CH—8172 ± 1003	
					Analysis: National League vs. All-Ireland Championship	TD (m·min^−1^)	NL—106 ± 17	
						CH—110 ± 14	
					HSR (m) (≥17 km·h^−1^)	NL—1215 ± 369	
							CH—1253 ± 258	
						HSR (m·min^−1^) (≥17 km·h^−1^)	NL—17 ± 5	
							CH—18 ± 4	
						SD (m) (≥22 km·h^−1^)	NL—362 ± 127	
							CH—406 ± 86	
						SD (m·min^−1^) (≥22 km·h^−1^)	NL—5 ± 2	
							CH—6 ± 1	
						Max Velocity (km·h^−1^)	NL—30.0 ± 1.7	
							CH—31.3 ± 1.2

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
