# Peer review of "Science and Hurling: A Review"

_sports, 2022, doi:10.3390/sports10080118_

Round 1

Reviewer 1 Report

The work presented a game little known in the wide world.

Thanks to the authors for an interesting review. This work can help develop the "hurling" like physical activity.

The work requires repairs:

1. It is necessary to formulate the purpose of the work separately in the introduction.

2. The introduction should list the practical and scientific aspects of the benefits of the work, which need to be elaborated on later.

3. In the conclusion it is necessary to present the main results of the work and recommendations for preparation.

Sincerely.

Author Response

I thank the reviewer for their direction. Please see below how the below points have been addressed.

1. It is necessary to formulate the purpose of the work separately in the introduction.

Comment: A specific aims been inputted in line 78 and 79.

2. The introduction should list the practical and scientific aspects of the benefits of the work, which need to be elaborated on later.

Comment: The introduction has been enhanced, this is specifically outlined in the addition of a new paragraph between line 68 and 79.

3. In the conclusion it is necessary to present the main results of the work and recommendations for preparation.

The conclusion has been enhanced with the addition of an additional paragraph and can be observed between line 605 and 620. 

Sincerely.

Reviewer 2 Report

Dear authors,

Your paper seems well structured and of scientific interest, but some aspects have to be addressed.

A merit of this review is that it offers an injury analysis for hurling in a specific section. Nevertheless, it should be improved in order to propose injury prevention and rehabilitation strategies. To do that making a comparison with other sports, I suggest the following references:

Farì G, Notarnicola A, DI Paolo S, Covelli I, Moretti B. Epidemiology of injuries in water board sports: trauma versus overuse injury. J Sports Med Phys Fitness. 2021 May;61(5):707-711. doi: 10.23736/S0022-4707.20.11379-3. PMID: 33975428.

- Farì G, Santagati D, Macchiarola D, Ricci V, Di Paolo S, Caforio L, Invernizzi M, Notarnicola A, Megna M, Ranieri M. Musculoskeletal pain related to surfing practice: Which role for sports rehabilitation strategies? A cross-sectional study. J Back Musculoskelet Rehabil. 2022 Jan 14. doi: 10.3233/BMR-210191. Epub ahead of print. PMID: 35068441.

Best regards

Author Response

I thank the reviewer of their feedback. The identified references have now been added to the review.

Round 2

Reviewer 1 Report

Thanks to the authors for the exciting research.

I think the manuscript can be published in the last form.

Sincerely.